# Heparanase (HPSE) Associates with the Tumor Immune Microenvironment in Colorectal Cancer

Mengling Liu [1,†], Qing Liu [1,2,†], Yitao Yuan [1], Suyao Li [1], Yu Dong [1], Li Liang [1], Zhiguo Zou [3,*] and Tianshu Liu [1,2,*]

1   Department of Medical Oncology, Zhongshan Hospital, Fudan University, Shanghai 200032, China; liuml13@fudan.edu.cn (M.L.); liu.qing@zs-hospital.sh.cn (Q.L.); yuanyitao1997@163.com (Y.Y.); lisuyao0214@163.com (S.L.); dongyu3664@163.com (Y.D.); liang.li@zs-hospital.sh.cn (L.L.)
2   Cancer Center, Zhongshan Hospital, Fudan University, Shanghai 200032, China
3   Department of Cardiology, Renji Hospital, School of Medicine, Shanghai Jiaotong University, Shanghai 200127, China
*   Correspondence: zouzhiguo@renji.com (Z.Z.); liu.tianshu@zs-hospital.sh.cn (T.L.)
†   Authors with equal contribution.

**Abstract:** There is an unmet clinical need to identify potential predictive biomarkers for immunotherapy efficacy in mismatch repair proficient (pMMR) metastatic colorectal cancer (mCRC). Heparanase (HPSE) is a multifunctional molecule mediating tumor–host crosstalk. However, the function of HPSE in the tumor immune microenvironment of CRC remains unclear. Data of CRC patients from public datasets (TCGA and GSE39582) and Zhongshan Hospital (ZS cohort) were collected to perform integrative bioinformatic analyses. In total, 1036 samples from TCGA (N = 457), GSE39582 (N = 510) and ZS cohort (N = 69) were included. Samples of deficient MMR (dMMR) and consensus molecular subtypes 1 (CMS1) showed significantly higher HPSE expression. The expression of HPSE also exhibited a significantly positive association with PD-L1 expression, tumor mutation burden and the infiltration of macrophages. Immune pathways were remarkably enriched in the HPSE high-expression group, which also showed higher expressions of chemokines and immune checkpoint genes. Survival analysis suggested that high HPSE expression tended to be associated with shorter overall survival in patients with pMMR mCRC. HPSE might contribute to the immune-activated tumor microenvironment with high levels of immune checkpoint molecules, suggesting that pMMR mCRC with high HPSE expression might respond to immune checkpoint inhibitors.

**Keywords:** HPSE; colorectal cancer; tumor microenvironment; mismatch repair proficiency

## 1. Introduction

Colorectal cancer (CRC) is one of the critical causes of cancer-related mortality worldwide [1]. Despite unremitting efforts devoted to finding the optimal management of colorectal cancer, the prognosis for patients with metastatic CRC (mCRC) remains poor [2]. During recent years, immunotherapy has dramatically reformed the cancer therapeutic strategy. Biomarkers of response to immunotherapy have been explored widely in cancers. Tumor mutation burden (TMB) [3], tumor PD-L1 expression [4,5], and immune cells infiltration in the tumor microenvironment (TME) [6] are all important biomarkers of the immune checkpoint inhibitors (ICIs) response, but none of these alone seem to be sufficient for predicting immunotherapy efficiency in CRC [7,8]. More precise and reliable biomarker are needed to be identified for ICIs therapy in CRC.

Consensus molecular subtypes (CMS) and mismatch repair (MMR) subtypes are robust molecular classifications in CRC [9]. Four CMS groups (CMS1-4) provide the best current description of CRC heterogeneity at the transcriptomic level, while subgroups with different MMR status display disparate mutational profiles. The CMS1 group is highly enriched in microsatellite instability (MSI) tumors with hypermutation, hypermethylation, and a strong infiltration of the TME with immune cells [10]. MMR deficiency (dMMR)

causes MSI in tumors due to the deficient activity in the surveillance and correction of errors during DNA replication, repair, and recombination [11]. Tumors with dMMR are characterized by high TMB and heavy immune cell infiltrations [12], similar to the CMS1 group. The presence of dMMR in CRC has been a distinct biomarker for the potential response to ICIs therapy, but efficient predictive biomarkers are absent in mCRC with mismatch repair proficiency (pMMR), which is the most common form among mCRC patients.

Heparanase (HPSE) is a unique mammalian endo-β-D-endoglycosidase that cleaves heparan sulphate, an important component of the extracellular matrix. This leads to the remodeling of the extracellular matrix, whilst liberating growth factors and cytokines bound to heparan sulphate. This in turn promotes both physiological and pathological processes such as angiogenesis, immune cell migration, inflammation, wound healing, and metastasis [13,14]. Furthermore, HPSE exhibits non-enzymatic actions in cell signaling and in the regulation of gene expression [13,14]. It has been reported that HPSE promotes an immunosuppressive TME by regulating the activation of macrophages [15,16] and mediates tumor immunosurveillance via natural killer (NK) cells [17]. HPSE was also shown to regulate the secretion of cytokines to establish a chemokine gradient and facilitate immune cell recruitment [15].

Given the important role of HPSE and the TME in cancer, we intended to examine whether there were critical associations between tumor HPSE expression and the immune microenvironment in CRC. We further investigated the prognostic role of HPSE in pMMR mCRC, providing some insights into the potential role of HPSE expression as a predictive biomarker of ICIs response.

## 2. Materials and Methods

### 2.1. Public Datasets and Clinical Samples

Public transcriptome and clinical data of CRC tissue samples were downloaded from The Cancer Genome Atlas (TCGA) data portal (https://portal.gdc.cancer.gov/, accessed date: 2 October 2019) [18] and GSE39582 dataset (https://www.ncbi.nlm.nih.gov/geo/query/acc.cgi?acc=GSE39582, accessed date: 2 October 2019) [19]. Samples with missing values of MMR or KRAS/BRAF mutation status were excluded. Genome-wide mutation data of TCGA cohort were also downloaded to calculate TMB, defined as the total number of coding mutations per megabase. No mutation data of whole genome for GSE39582 cohort was available. Tissue samples and clinical records of 69 patients diagnosed with pMMR CRC at Zhongshan Hospital Fudan University were collected as ZS cohort, which was approved by the Ethics Committee of Zhongshan Hospital Fudan University. Written informed consent was obtained from all participants. The HPSE expression of the ZS cohort was evaluated by immunohistochemical staining (IHC). All patients with pMMR mCRC included in the study received non-immunotherapy treatment.

### 2.2. Immune Cell Infiltration Evaluation

Normalized gene expression data of TCGA and GSE39582 were uploaded on CIBER-SORTx (https://cibersortx.stanford.edu/, accessed date: 8 May 2020) [20] and xCell (https://xcell.ucsf.edu/, accessed date: 8 May 2020) [21]. Immune cell scores were computed by LM22 gene signatures at CIBERSORTx with recommended parameters (Job type: Impute Cell Fractions; Batch correction: disabled; Disable quantile normalization: true; Run mode: relative, Permutations: 100) and by xCell gene signatures of 64 immune and stroma cell types. IHC for four immune cell markers was performed on tissue samples of ZS cohort to assess the infiltration of immune cells, including CD4+ T cell, CD8+ T cell, CD19+ B cell, and CD68+ macrophages.

### 2.3. Immunohistochemical Staining (IHC)

IHC was conducted according to the manufacturer's instructions. The following antibodies were applied: HPSE antibody (24529-1-AP, 1:100, Proteintech Group, Wuhan, Hubei,

China), CD4 antibody (GB13064, 1:100, Servicebio Technology, Wuhan, Hubei, China), CD8 antibody (GB13068, 1:100, Servicebio Technology), CD19 antibody (GB11061, 1:500, Servicebio Technology) and CD68 antibody (GB13067-M-2, 1:100, Servicebio Technology). H-Score was calculated to evaluate the expression of each marker by the Quant Center 2.1 (3DHISTECH, Budapest, Hungary) using the following formula: H-Score = (percentage of cells of weak intensity *1) + (percentage of cells of moderate intensity *2) + (percentage of cells of strong intensity *3).

### 2.4. Gene Set Enrichment Analysis (GSEA)

Samples of TCGA and GSE39582 were divided into two groups according to the transcriptional level of HPSE, with the median as the cutoff value. Using "all GO gene sets" and "KEGG gene sets" downloaded from Molecular Signatures Database (v7.1), GSEA was performed to identify the biological pathways that differed between high and low HPSE expression groups by R packages. Terms were selected from the top five pathways according to *p*-values.

### 2.5. Statistical Analysis

The Wilcoxon rank sum test and the Kruskal–Wallis test were used to compare differences between two groups and multiple groups respectively. Spearman´s correlation was applied to all correlation analyses. Survival analysis was conducted by the Kaplan-Meier survival curve with Log-rank test. Hazard ratio (HR) and its 95% confidence interval (CI) were calculated by the Cox proportional hazard model. Values of $p < 0.05$ were considered statistically significant. All statistical analyses and figure drawings were completed in R (version 3.6.3).

### 3. Results

#### 3.1. HPSE Expression in Different Molecular Subtypes of CRC and Its Association with PD-L1 Expression and TMB

We first compared the HPSE expression level in different MMR and CMS subgroups using the transcriptional data of 967 samples from TCGA (N = 457) and GSE39582 (N = 510) datasets (Figure 1, Table 1). Samples of dMMR subgroup and CMS1 subgroup showed significantly higher HPSE expressions at the transcriptional level (Figure 2a,b) in TCGA and GSE39582. Next, we explored the correlation between HPSE and PD-L1 expression, which revealed a strong positive correlation in both datasets (Figure 2c). The positive association of HPSE expression and TMB was also observed in TCGA (Figure 2d). These results suggested a possible function of HPSE in modulating the immune profile of CRC.

#### 3.2. Higher HPSE Expression Associated with an Increased Infiltration of Immune Cells

To investigate whether HPSE promotes the infiltration of immune cells, gene expression data from TCGA and GSE39582 were analyzed using CIBERSORTx [20] to estimate the abundance of 22 types of immune cells and xCell [21] to compute the immune score and stroma score based on the enrichments of 64 immune and stroma cell types within each sample. Spearman's correlation analysis showed a significant positive correlation between the HPSE expression level and the infiltration of activated NK cells, M1 macrophages and neutrophils, but no clear associations were observed for T cells and B cells (Figure 3). The immune score and microenvironment score were strongly correlated with HPSE expression, which was consistent with PD-L1 and interferon gamma (IFNG), the inducer of PD-L1 transcription [22] (Figure 3). To validate these results, we used a tissue microarray of 69 samples (ZS cohort, Table 1) with pMMR CRC to assess the HPSE expression and the infiltration of four immune cells (CD4+ T cells, CD8+ T cells, CD19+ B cells and CD68+ macrophages) by IHC. A lack of correlation was found between HPSE expression and CD4+ T cells (Figure 4b), and CD19+ B cells (Figure 4c) infiltration. However, a robust positive association between HPSE expression and CD8+ T cells (Figure 4d) and macrophages

infiltration (Figure 4e) was observed. These data indicated that HPSE might promote the recruitment of immune cells, especially macrophages in pMMR CRC.

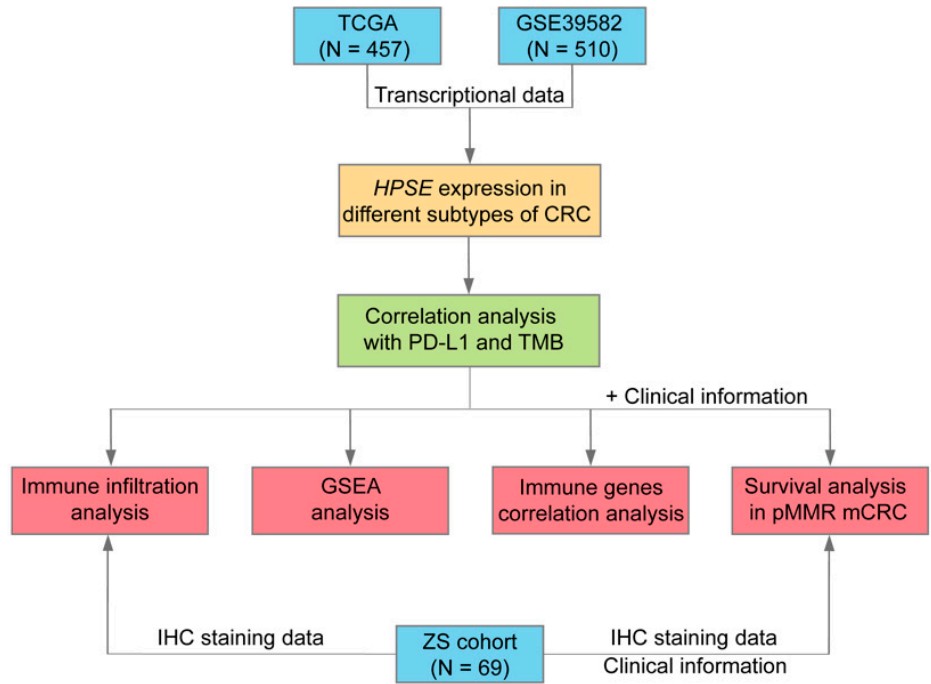

**Figure 1.** The flowchart of this study. CRC, colorectal cancer; TMB, tumor mutation burden; GSEA, Gene set enrichment analysis; pMMR, mismatch repair-proficient; mCRC, metastatic colorectal cancer; IHC, immunohistochemical staining.

**Table 1.** The clinical characteristics of patients with CRC in TCGA, GSE39582, and the ZS cohort.

| | | Number of Patients | | |
|---|---|---|---|---|
| | | **TCGA** | **GSE39582** | **ZS Cohort** |
| Total | | 457 | 510 | 69 |
| Gender | Female | 213 | 233 | 23 |
| | Male | 241 | 277 | 46 |
| Age | Mean (SD) | 66.4 (12.7) | 66.9 (13.1) | 60.1 (10.25) |
| Site | Left | 223 | 303 | 46 |
| | Right | 141 | 207 | 23 |
| CMS | CMS1 | 45 | 86 | - |
| | CMS2 | 175 | 204 | - |
| | CMS3 | 47 | 65 | - |
| | CMS4 | 101 | 113 | - |
| MMR | dMMR | 52 | 72 | 0 |
| | pMMR | 403 | 391 | 69 |
| BRAF | Mutant | 31 | 51 | 3 |
| | Wildtype | 426 | 459 | 66 |
| KRAS | Mutant | 123 | 204 | 37 |
| | Wildtype | 334 | 306 | 32 |
| Metastasis | M0 | 365 | 446 | 29 |
| | M1 | 70 | 60 | 40 |

SD, standard deviation; CMS, consensus molecular subtype; MMR, mismatch repair; dMMR, mismatch repair deficiency; pMMR, mismatch repair proficiency.

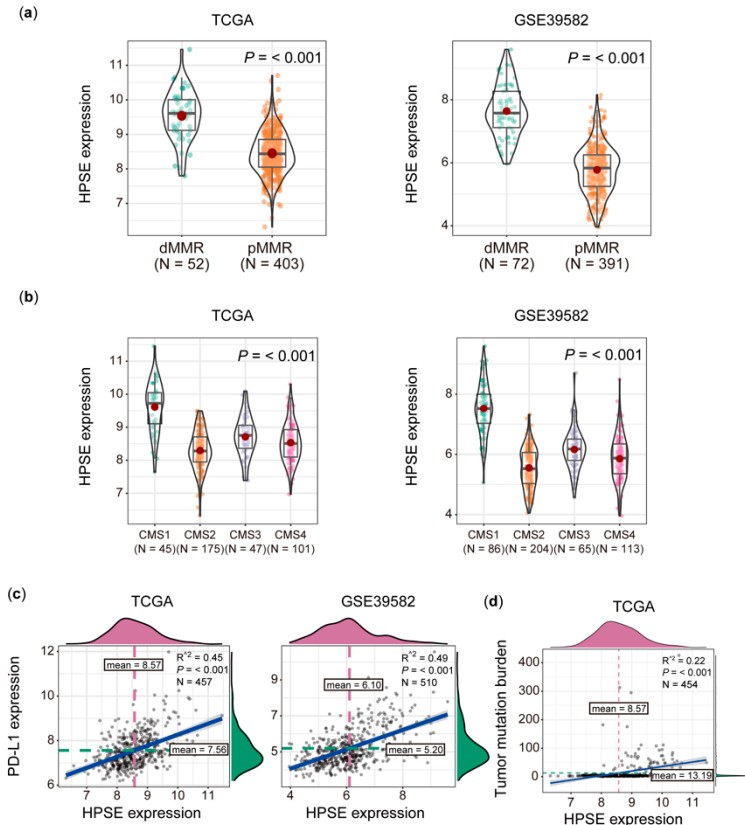

**Figure 2.** HPSE expression in different molecular subtypes of CRC and its association with PD-L1 expression and tumor mutation burden (TMB). (**a**) HPSE expression in CRC with different MMR status. (**b**) HPSE expression in CRC with different CMS subtypes. (**c**) Correlation analysis between expressions of HPSE and PD-L1. (**d**) Correlation analysis between HPSE expression and TMB. MMR, mismatch repair; dMMR, mismatch repair deficiency; pMMR, mismatch repair proficiency; CMS, consensus molecular subtype.

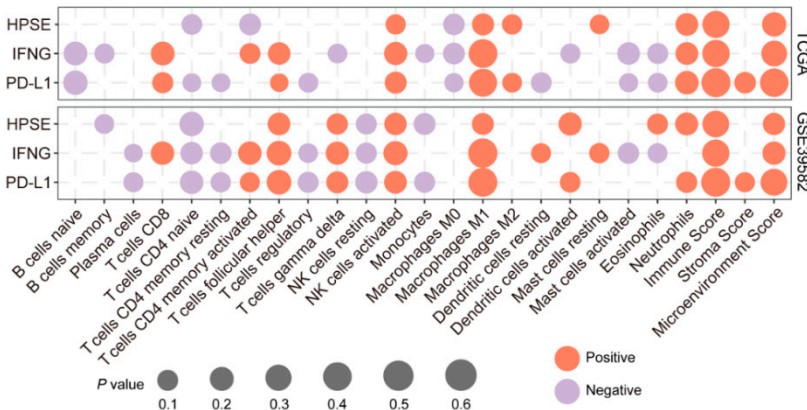

**Figure 3.** The heatmap of the correlations between gene expressions and immune cells infiltration in TCGA and GSE39582.

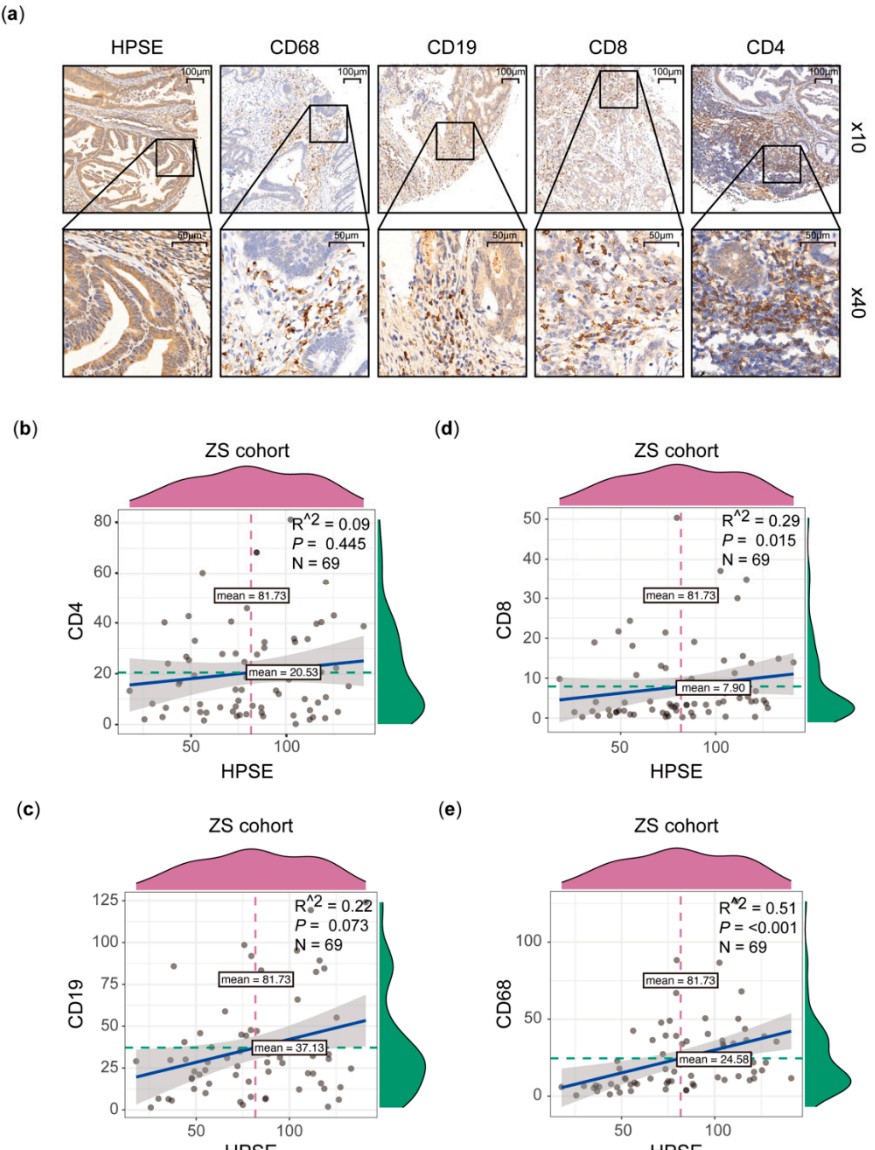

**Figure 4.** Correlation analyses of HPSE expression and the immune cells infiltration in ZS cohort. (**a**) Representative images of immunohistochemistry staining. (**b**) The correlation between HPSE expression and CD4+ T cell infiltration. (**c**) The correlation between HPSE expression and CD19+ B cell infiltration. (**d**) The correlation between HPSE expression and CD8+ T cell infiltration. (**e**) The correlation between HPSE expression and CD68+ macrophages infiltration.

### 3.3. Immune Pathways Were Enriched in the HPSE Expression-High Group

GSEA was performed to identify the Gene Ontology (GO) pathways and the Kyoto Encyclopedia of Genes and Genomes (KEGG) that differ between high and low HPSE expression groups. The top five differential KEGG and GO pathways in the TCGA and GSE39582 datasets were presented in Figure 5. Immune pathways were remarkably enriched in the HPSE expression-high group, among which "POSITIVE REGULATION OF DEFENSE RESPONSE" (Figure 5a), "ADAPTIVE IMMUNE RESPONSE" (Figure 5b), and "CYTOKINE-CYTOKINE RECEPTOR INTERACTION" (Figure 5c,d) were the most significantly enriched pathways, which supports the contribution of HPSE to the immune-activated microenvironment in CRC.

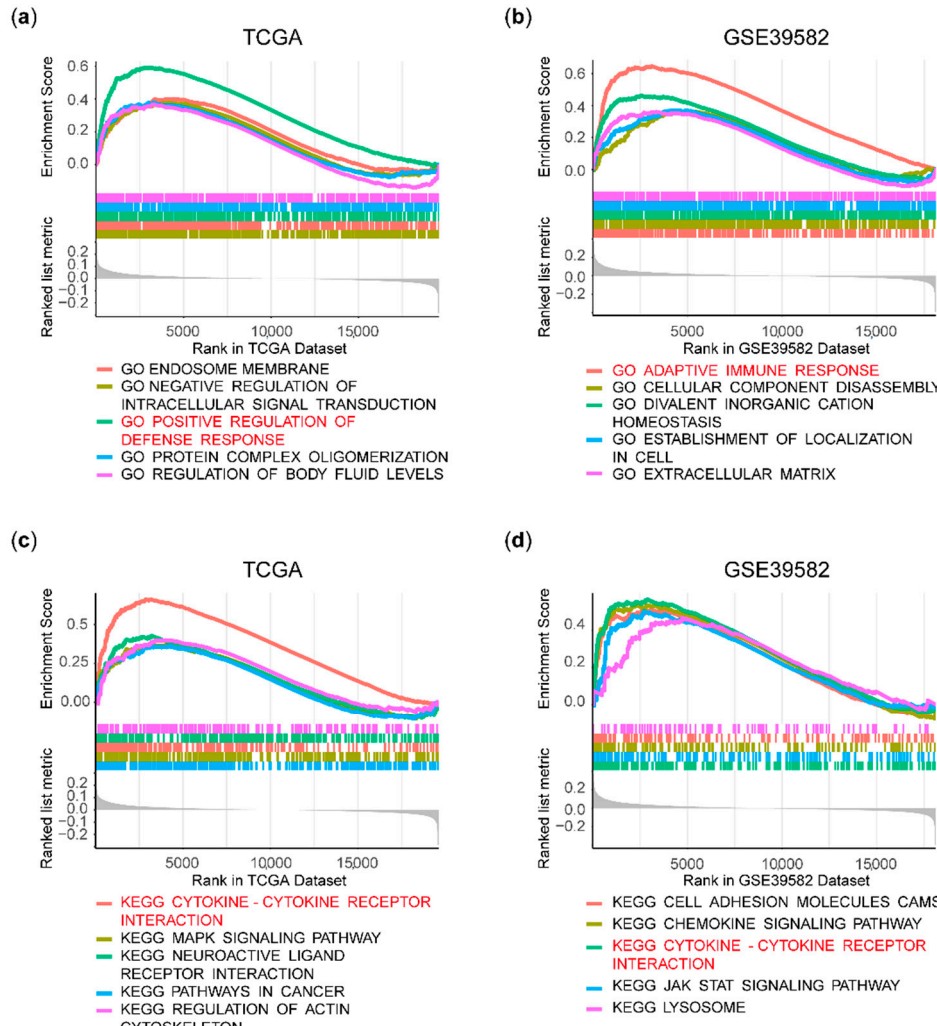

**Figure 5.** Gene set enrichment analysis based on GO terms (**a**,**b**) and KEGG terms (**c**,**d**) in TCGA and GSE39582. The top enriched terms were colored in red. GO, Gene Ontology; KEGG, Kyoto Encyclopedia of Genes and Genomes.

### 3.4. Correlations between HPSE Expression and Immune Genes

Chemokines regulate the function of immune cells in the TME to promote or inhibit tumor progression [23]. We identified statistically significant positive or negative associations between expressions of HPSE and genes of the chemokine family, such as CCL4, CCL13, and CCR4 (Figure 6a), which showed a strong positive correlation with HPSE expression. The increased expression of immune checkpoint proteins such as PD-1 and CTLA-4 was closely related to the failure of immunosurveillance [24]. The idea that HPSE associated with the expression of immune checkpoint genes provoked our interest. We analyzed the expressions of 47 immune checkpoint genes and found that 33 of them had a statistically significant correlation with HPSE expression both in TGCA and GSE39582 datasets, among which 90% were positively correlated (Figure 6b). We then focused on the pMMR subpopulation and identified 24 genes with a positive association with HPSE expression, especially CD28 and CD274 (namely PD-L1) (Figure 6c).

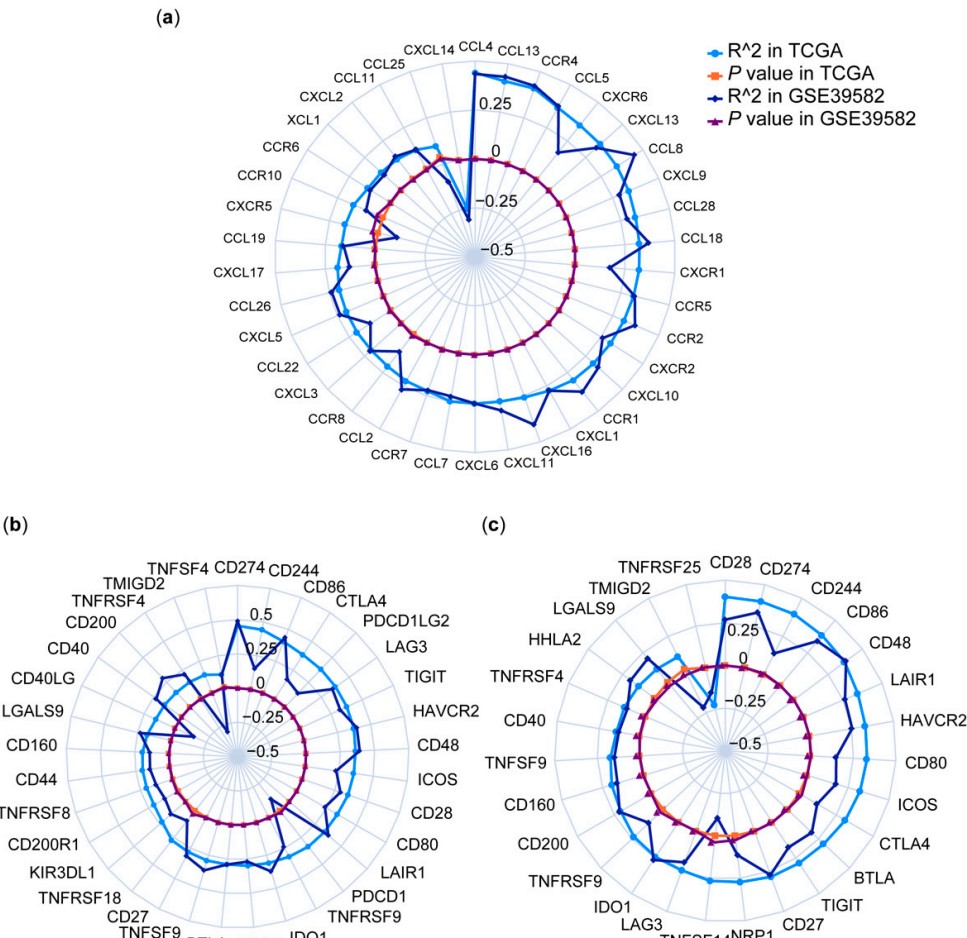

**Figure 6.** Correlation analyses of HPSE and immune genes expression in CRC. (**a**). Correlation between expression of HPSE and chemokine family. (**b**) Correlation between expression of HPSE and immune checkpoint genes. (**c**) Correlation between expression of HPSE and immune checkpoint genes in the pMMR subgroup. Genes with $p < 0.05$ were shown in the radars.

*3.5. Poor Survival in pMMR mCRC Patients with High HPSE Expression*

Survival analysis was performed in a stage IV pMMR CRC subpopulation of TCGA, GSE39582, and ZS cohorts to investigate the prognostic role of HPSE. HPSE expression at the transcriptional level did not associate with the survival of pMMR CRC in TCGA (Figure 7a), but patients with high HPSE expression showed a significantly inferior overall survival (OS) in GSE39582 (Figure 7b). In the ZS cohort, a poorer tendency of survival was observed in the high HPSE expression group, with a median OS (mOS) of 615 days, whereas the low HPSE expression group had for a mOS of 2310 days. However, the difference was not statistically significant (Figure 7c).

**4. Discussion**

The identification of the population in pMMR CRC that might benefit from ICIs therapy remains a major clinical challenge [7]. HPSE, as a multifunctional molecule and a key regulator of major TME components, including cancer cells, immune cells, cancer-associated fibroblasts, endothelial cells, and pericytes, has been reported in various types of cancer [14]. To investigate the potential for HPSE to act as a predictive biomarker for immunotherapy, we studied the correlation between HPSE and the immune profile of CRC by bioinformatic analyses using public datasets and our own clinical cohort.

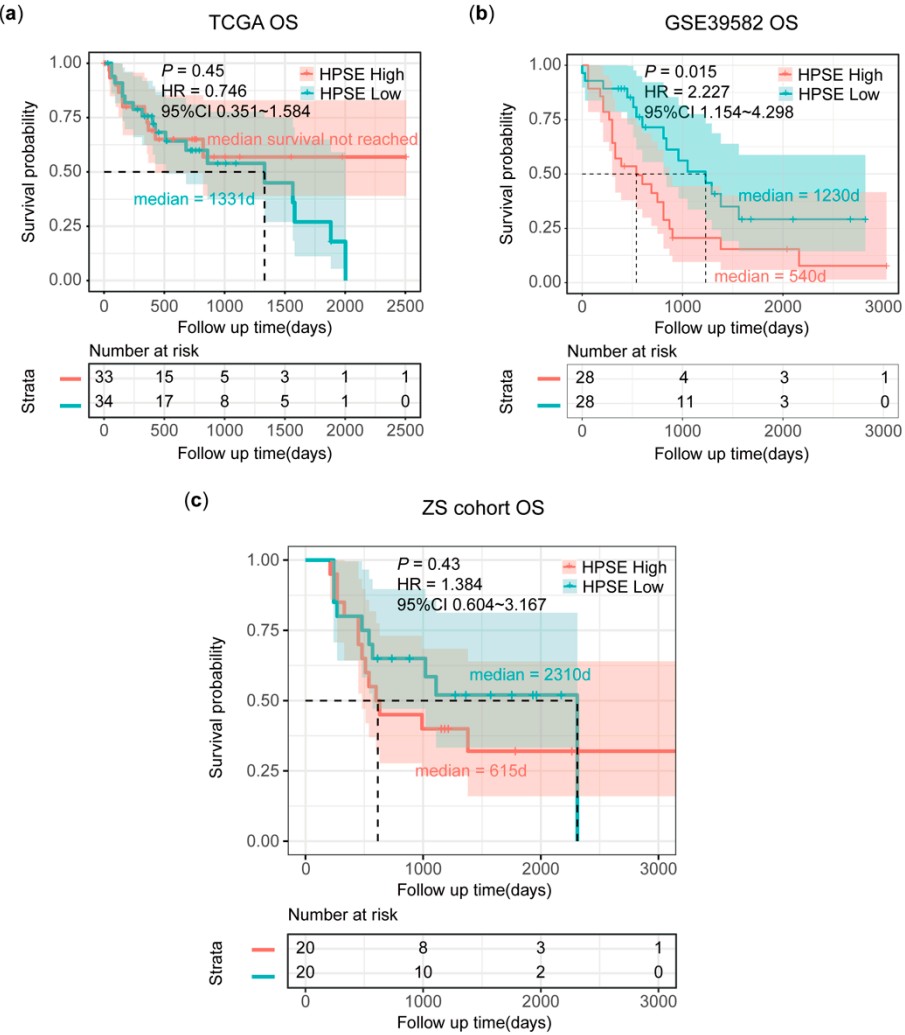

**Figure 7.** Kaplan–Meier survival curves for pMMR mCRC in TCGA (**a**), GSE39582 (**b**), and ZS (**c**) cohorts. OS, overall survival.

We firstly found that patients of immune-activated groups (the dMMR and CMS1 groups) presented a significantly higher level of HPSE expression. Tumors with dMMR or CMS1 were characterized by hypermutation and the strong infiltration of immune cells in the TME [10], which responded well to ICIs therapy [7]. We further analyzed the correlation between the expression of HPSE and two widely explored biomarkers of immunotherapy efficacy, PD-L1 [4] and TMB [3], and revealed a strong positive association based on the two public datasets. Therefore, we suspected that HPSE expression might be a potential predictive biomarker for ICIs therapy in CRC.

It has been reported that HPSE is pivotal in the activation and function of macrophages in the TME [15] and mediates the recruitment of neutrophils and lymphocytes by regulating the secretion of chemokines [25–28]. In our study, immune cells infiltration evaluation showed that HPSE expression displayed a remarkably positive correlation with the infiltration level of activated NK cells, M1 macrophages and neutrophils in both TCGA and GSE39582 datasets. Results from the pMMR CRC subgroup also revealed that samples with a higher HPSE expression exhibited increased infiltration of CD8+ T cells and macrophages. Activated NK cells, neutrophils, M1 macrophages and CD8+ T cells were all key players in tumor immune surveillance [24,29,30]. No strong evidence in our study indicated that M2 macrophage and tumor-associated neutrophil were associated with HPSE expression. Therefore, our analysis suggests that high HPSE expression might promote immune surveillance in CRC. Immune pathways enrichment and gene expression analyses also indicated

the critical role of HPSE in immune activation. Gutter-Kapon et al. reported that HPSE was shown to regulate the secretion of cytokines, such as TNF-α, IL-1β, IL-10 and IL-6 to establish a chemokine gradient and facilitate immune cell recruitment [15]. In a recent landmark study, an increase in HPSE activity in CAR-T cells was shown to significantly enhance tumor invasion and anti-tumor immunity [31]. Therefore, HPSE expression may promote the infiltration of immune cells in CRC.

CCL4, also known as macrophage inflammatory protein-1 (MIP-1β), plays a key role in cancer progression by mediating the interaction between cancer cells and fibroblasts [32], or by inducing vascular endothelial growth factor expression and lymphangiogenesis [33]. The importance of CCL13, also known as monocyte chemoattractant protein (MCP-4), has never been reported in cancer yet [34], despite the observation that it may increase apoptosis resistance in other diseases [35]. CCR4 is the receptor of CCL2, CCL17 and CCL22. Their pro-tumor and anti-tumor effects, caused by the recruiting of different types of immune cells have been well studied [34]. Strong positive correlations between the expressions of HPSE and CCL4, CCL13, and CCR4, as well as other chemokine family genes, further suggested the complicated role of HPSE in the TME of CRC.

CD28 is the essential costimulatory molecule in T cell activation. A recent study revealed that PD-1 suppressed the T cell function primarily by inactivating CD28 signaling, suggesting that costimulatory pathways played key roles in regulating the effector T cell function and responses to anti–PD-L1/PD-1 therapy [36]. In the pMMR CRC subgroup, we discovered a distinct association between expressions of HPSE and CD28, as well as PD-L1, which supported HPSE as a potential predictive biomarker of ICIs treatment response for patients with pMMR CRC.

More importantly, although immune infiltration analysis suggested that high HPSE expression might promote the immune activation in CRC, genes expression analysis indicated that these tumors may escape from immune surveillance by expressing immune checkpoint molecules, similar to the dMMR group and CMS1 group [7,9]. This also explained the result that stage IV pMMR patients with non-immunotherapy treatment in the high HPSE expression group still tended to have shorter OS in the GSE39582 and ZS cohorts.

## 5. Conclusions

HPSE might contribute to the poor survival of pMMR mCRC patients by enhancing the immune escape from surveillance, and it can be a biomarker for a potential response to ICIs therapy in this subpopulation. However, this study only provides some preliminary clues, and future laboratory and clinical studies are warranted.

**Author Contributions:** Conceptualization, M.L. and Q.L.; methodology, M.L.; software, M.L.; validation, M.L. and Q.L.; formal analysis, M.L. and Y.Y.; investigation, M.L. and S.L.; resources, Q.L. and L.L.; data curation, M.L. and Y.Y.; writing—original draft preparation, M.L., Q.L. and Y.D.; writing—review and editing, Q.L. and Z.Z.; visualization, M.L.; supervision, Z.Z. and T.L.; project administration, Z.Z. and T.L.; funding acquisition, Q.L. and T.L. All authors have read and agreed to the published version of the manuscript.

**Funding:** This research was funded by the National Natural Science Foundation of China, grant number 81772511 and 81802356.

**Institutional Review Board Statement:** The study was conducted according to the guidelines of the Declaration of Helsinki, and approved by the Institutional Review Board (or Ethics Committee) of Zhongshan Hospital Fudan University (protocol code B2017-165R, date of approval: 15 February 2019).

**Informed Consent Statement:** Informed consent was obtained from all subjects involved in the study.

**Data Availability Statement:** The data presented in this study are available on request from the corresponding authors.

**Conflicts of Interest:** The authors declare no conflict of interest. The funders had no role in the design of the study; in the collection, analyses, or interpretation of data; in the writing of the manuscript, or in the decision to publish the results.

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
