# Peer review of "Heparanase (HPSE) Associates with the Tumor Immune Microenvironment in Colorectal Cancer"

_processes, doi:10.3390/pr9091605_

Round 1
Reviewer 1 Report
The manuscript by Mengling Liu et al. entitled “ HPSE is a potential predictive biomarker for immunotherapy in colorectal cancer”intends to demonstrate that Heparanase can be used as a potential biomarker for immunotherapy in colorectal cancer patients. In particular, the authors have utilized data of CRC patients from public datasets (TCGA and GSE39582) and Zhongshan Hospital (ZS cohort) to perform integrative bioinformatic analyses. Samples of deficient MMR (dMMR) and CMS1 subgroups showed significantly higher HPSE expression. Expression of HPSE also exhibited signif- icantly positive association with PD-L1 expression, tumor mutation burden and the infiltration ofmacrophages. Survival analysis suggested that high HPSE expression was an indicator of poor prognosis in patients with pMMR mCRC. HPSE might contribute to the immune-activated tumor microenvironment with high levels of immune checkpoint molecules, suggesting that pMMR mCRC with high HPSE expression might respond to immune checkpoint inhibitors.
The experimental approach is more than adequate and the conclusions at which they arrive seem to be appropriate. This work is noteworthy as it aims to demonstrate for the first time that Hpse not only facilitates cell invasion and metastasis but also contributes to pro-tumorigenic and pro-metastatic interactions between the tumor cells and the host tumor microenvironment, immune cells and systemic soluble factors.
Nevertheless the authors should address the following points prior to publication :
1) A short list of abbreviations should be provided to allow for easy reading of the text
2) In the introduction, the part concerning the role of the HPSE is too short (from line 53 to line58). It should be supplemented with more information about the enzymatic and non-enzymatic role of this enzyme in order to help the less experienced reader of HPSE. It should also be remembered the role played by HPSE in conditioning the tumor microenvironment (TME) and in promoting inflammation and the response of immune cells. To this end, other citations should be included (eg Masola V. et al. Role of heparanase in tumor progression: Molecular aspects and therapeutic options. Semin Cancer Biol. 2020)
Author Response
Comment 1:
Thanks for the kind reminder, and we have updated the list of abbreviations in our revised manuscript to allow for easier reading.
Comment 2:
More information regarding the enzymatic and non-enzymatic role of HPSE has been added as suggested in the introduction (line 53-62). We also appreciate the recommendation of the comprehensive review, which has promoted a better understanding of HPSE in the research area of cancer. The review has been cited as our Reference 13.

Reviewer 2 Report
In this manuscript, the authors assessed transcriptome data of two public databases and immunohistochemistry data of their own cohort, and described high HPSE expression in the dMMR and CMS1 subtypes of CRC, correlation of HPSE expression and PD-L1 expression and TMB, correlation of HPSE expression and infiltration of immune cells especially macrophages, etc. They also assessed the correlation of HPSE expression and overall survival of pMMR mCRC patients and one of three cohorts revealed statistically significant poor prognosis in HPSE High patients.
As the authors described in the conclusion, this is a preliminary study of correlation analysis basically using public database. I raised several points below.
1) The authors showed positive correlation of HPSE expression and infiltration of immune cells in Figure 3a by transcriptome data and tried to validate it by immunohistochemistry in a cohort of pMMR CRC in Figure 3b-e.
1-1) Representative photo images of immunohistochemistry of each protein examined are needed. Especially, is HPSE expressed in the cancer cells? What cell is the main source of HPSE?
1-2) They examined only CD4, CD8, CD19, CD68 and HPSE, but markers of neutrophil, NK cell and M1 (and M2) macrophage were not examined. It seems worth staining them.
1-3) PD-L1 expression should be examined by immunohistochemistry.
1-4) They showed positive correlation of HPSE expression and immune cell markers expression. Some of them are known to activate immune surveillance, but some are reported to be associated with immunosuppressive or tumor-promoting microenvironment (for example, M2 macrophage and tumor-associated neutrophil etc.), therefore, the function of HPSE as a whole remains unclear. The authors need to examine some more immune cell markers and discuss about this.
1-5) This study only showed correlation of HPSE expression and immune cell markers expression. Whether HPSE expression really caused increase of those immune cell markers expression, or immune cell infiltration caused HPSE expression remains unclear. Can the authors discuss about such a mechanistic explanation?
2) In line 202-3, the authors described, “the difference was not statistically significant due to the small number of patients enrolled in our cohort”, but the number of patients was not so different between Figure 6a-c and Figure 6b showed statistical difference, therefore, the explanation is not appropriate. Survival analysis in this study showed no significant difference in the two cohorts among the three, therefore, cannot conclude that high HPSE expression is an indicator of poor prognosis.
3) In line 146, the authors described, “no obvious associations were observed for T cells and B cells”, which seems to mean “cold tumor”. The authors repeatedly described that HPSE can be a potential predictive biomarker for immunotherapy in CRC, but it is just a speculation and is there any discussion whether anti-PD-1/PD-L1 therapy really effective for such a “cold tumor”?
4) Abbreviations should be spelled out for the first appearance: CMS1 in the Abstract, IFNG in line 148.
5) Figures should be labelled in order of appearance: Figure 3c and 3d in lines 153-4.
Author Response
Comment 1-1
Representative images of immunohistochemistry of each protein examined have been added in Figure 4A. As indicated in the figure, HPSE was mainly expressed in nuclei and cytoplasm of cancer cells.
Comment 1-2 and 1-3
We agree with the reviewer on this important point, however, we did not keep enough samples to complete staining for those important markers. At the moment, we are trying to collect tissue samples and complete the IHC analysis as required. Due to COVID-19 related issues, the work was not done before the due date.
Comment 1-4
We totally understand the reviewer’s concerns. As elaborated in the discussion (Line 234), in our study, immune cells infiltration evaluation showed that HPSE expression displayed remarkable positive correlation with the infiltration level of activated NK cells, M1 macrophages and neutrophils in both TCGA and GSE39582 datasets. In addition, results from the pMMR CRC subgroup revealed that samples with higher HPSE expression exhibited increased infiltration of CD8+ T cells and macrophages. It is well known that activated NK cells, neutrophils, M1 macrophages and CD8+ T cells are key players in tumor immune surveillance. On the other hand, there was no significant evidence in our study indicating that M2 macrophage and tumor-associated neutrophil were associated with HPSE expression. Immune pathways enrichment analysis also demonstrated a critical role of HPSE in immune activation. Therefore, our data suggested that high HPSE expression might promote immune surveillance in CRC. However, immune checkpoint genes expression analysis suggested that tumors with high HPSE expression may escape from immune surveillance with mechanisms involving immune checkpoint molecules, which was similarly observed in dMMR group and CMS1 group. To further address the reviewer’s concern, we are planning to perform more experiments (in vitro and in vivo) to investigate the exact role of HPSE in tumor development, which should be presented in our future study.
Comment 1-5
We understand the reviewer’s concern. As indicated in the introduction, HPSE exerts both enzymatic and non-enzymatic functions, contributing to a variety of physiological and pathological processes. In addition, Gutter-Kapon et.al reported that HPSE was shown to regulate the secretion of cytokines, such as TNF-α, IL-1β, IL-10 and IL-6 to establish a chemokine gradient and facilitate immune cell recruitment. In a recent landmark study, an increase in HPSE activity in CAR-T cells was shown to significantly enhance tumor invasion and anti-tumor immunity. Therefore, we carefully conclude that HPSE expression may promote the infiltration of immune cells in CRC. We understand that to fully address the reviewer’s concern on the relationship between HPSE and immune cell infiltration, more experiments are still needed, which should be the focus of our future study.
Comment 2
Thanks for the helpful suggestion. We are sorry for our inappropriate description and we’ve revised our conclusion accordingly in the revised manuscript (Line 24).
Comment 3
We understand the reviewer’s concern. Based on the gene expression data from TCGA and GSE39582, we did not observe obvious associations between T and B cells and HPSE expression, however, it might not be an indication of the absence of T & B cell infiltration in the tumor. Indeed, to assess the true protein expression level, we took advantage of IHC using a tissue microarray of our cohort with pMMR CRC, and a positive association between HPSE expression and CD8+ T cells (Figure 4d) and macrophages infiltration (Figure 4e) was found. To further address the reviewer’s concern regarding whether anti-PD-1/PD-L1 therapy really effective for a “cold tumor”, well-designed experiments are still required, which might be the focus of our future study.
Comment 4
Thanks for the kind reminder and we’ve revised our manuscript accordingly.
Comment 5
The manuscript has been revised accordingly, with figures labeled in the order of appearance.

Reviewer 3 Report
Liu et al. submitted an exciting paper showing the potential of heparinase expression as a biomarker in colorectal cancer therapy. The article is based on an advanced bioinformatic analysis of experimental data. The authors claimed that pMMR mCRC is an important goal because this CRC type lacks an efficient predictive biomarker for a potential response to ICIs therapy. The authors used available databases and tissue samples to assess the correlation between HPSE expression and the clinical outcome of patients. The manuscript is generally well written.
My primary concern is the lack of experimental group pf patients that received ICIs treatment. In the Materials and Methods section, the authors claimed that "All patients with pMMR mCRC included received non-immunotherapy treatment". It would be advantageous to compare the expression of HPSE in samples from ICI- and non-ICI therapy and then verify the correlation with therapy response in the context of HPSE. Without this type of analyzed group (ICIs-treated) it is quite speculatory to conclude about the importance of HPSE for immunotherapy, as stated in the title. Thus, in my opinion, the presented results do not support the manuscript hypothesis.
Minor comments:
1. Please use the whole name of HPSE in the title – Heparinase (HPSE)
2. Please, include the representative results of ICC analysis along with sample analysis.
Author Response
Primary concern
We fully understand the reviewer’s concern and agree with the reviewer on this important point. The involvement of CRC patients with ICI- and non-ICI therapy in the survival analysis will surely provide solid evidence further supporting our conclusion. Despite the great efforts we have made to find public CRC cohorts treated with ICIs, no applicable data could be acquired for our analysis at the moment. In fact, GSE179351 and GSE172162 datasets did not provide data regarding efficacy or survival. GSE136121 did not contain any gene expression data at baseline (before therapy), and MSKCC immunotherapy study provided only mutation data rather than information of mRNA and protein expression. Therefore, we carefully support a potential predictive role of HPSE expression for ICIs therapy in CRC, which is consistent with our description in the manuscript. We understand that further work with well-designed groupings including CRC patients receiving ICI and non-ICC treatment is still needed to elucidate the importance of HPSE for immunotherapy, which should be the focus of our future study.
Minor 1
Thanks for the kindly reminder, and we have corrected the error in our manuscript.
Minor 2
Representative images of immunohistochemistry for each examined protein have been added accordingly in Figure 4.

Round 2
Reviewer 2 Report
In this version, the authors added representative images of IHC and discussion about HPSE expression and immune activation.
This is a kind of preliminary study. To publish this study, at least PD-L1 immunohistochemistry might be needed. Or, correlation of PD-L1 and HPSE should be assessed using pMMR subgroups in the TCGA and GSE39582.
In lines 209-212, scientifically correct description is needed as “a poorer tendency of survival was also observed… the difference was not statistically significant possibly due to the small number of patients enrolled in our cohort.”
Author Response
We totally agree with that the correlation of PD-L1 and HPSE expressions in the pMMR subgroup is an important clue to support our conclusion. In fact, we have presented the associations between HPSE and 27 immune checkpoint genes including PD-L1 (CD274) in Figure 6c, which was analyzed using expression data of pMMR subgroups in TCGA and GSE39582. As indicated, PD-L1 showed a strong correlation with HPSE expression (R^2> 0.25, P < 0.001 both in TCGA and GSE39582) in the subgroup.
In addition, we’ve corrected our description in our revised manuscript accordingly (Line 211-215).
Thanks again for those constructive comments, and we have learned a lot from the review process, which should benefit our future study.
Reviewer 3 Report
I appreciate the corrected version of the manuscript and the Author's response. I understand that it isn't easy to obtain a statistically significant group of patients who receive ICI treatment. The authors agreed that well-designed groupings, including CRC patients receiving ICI and non-ICC treatment, is still needed to elucidate the importance of HPSE for immunotherapy, which should be the focus of our future study. Thus, maybe the title of the manuscript should be corrected and adjust more to the presented results? Without knowledge about HPSE expression in ICI-treated group, it is pretty speculatory whether it could be a biomarker for this kind of therapy?
Author Response
Thanks for the suggestion. We’ve corrected the title of our manuscript accordingly, which should be more suitable based on the data presented by this study.
